# Deep learning-based retinal vessel segmentation with cross-modal evaluation

**Luisa Sanchez Brea**               M.SANCHEZBREA@ERASMUSMC.NL
**Danilo Andrade De Jesus**          D.ANDRADEDEJESUS@ERASMUSMC.NL
**Stefan Klein**                    S.KLEIN@ERASMUSMC.NL
**Theo van Walsum**               T.VANWALSUM@ERASMUSMC.NL
*Biomedical Imaging Group Rotterdam, Department of Radiology and Nuclear Medicine,*
*Erasmus MC, Rotterdam, The Netherlands*

## Abstract

This work proposes a general pipeline for retinal vessel segmentation on *en-face* images. The main goal is to analyse if a model trained in one of two modalities, Fundus Photography (FP) or Scanning Laser Ophthalmoscopy (SLO), is transferable to the other modality accurately. This is motivated by the lack of development and data available in *en-face* imaging modalities other than FP. FP and SLO images of four and two publicly available datasets, respectively, were used. First, the current approaches were reviewed in order to define a basic pipeline for vessel segmentation. A state-of-art deep learning architecture (U-net) was used, and the effect of varying the patch size and number of patches was studied by training, validating, and testing on each dataset individually. Next, the model was trained in either FP or SLO images, using the available datasets for a given modality combined. Finally, the performance of each network was tested on the other modality. The models trained on each dataset showed a performance comparable to the state-of-the art and to the inter-rater reliability. Overall, the best performance was observed for the largest patch size (256) and the maximum number of overlapped images in each dataset, with a mean sensitivity, specificity, accuracy, and Dice score of $0.89\pm 0.05$, $0.95\pm0.02$, $0.95\pm0.02$, and $0.73\pm0.07$, respectively. Models trained and tested on the same modality presented a sensitivity, specificity, and accuracy equal or higher than 0.9. The validation on a different modality has shown significantly better sensitivity and Dice on those trained on FP.

**Keywords:** deep learning, retina, vessel segmentation, scanning laser ophthalmoscopy, fundus photography

## 1. Introduction

The eye is one of the most complex organs in the human body. Its importance is not just limited to the vision, as it also offers a possibility to non-invasively look at structures such as vessels. A healthy eye is composed of a sequence of refractive transparent structures that allow the light to be focused at the retina (Jesus and Iskander, 2015), where it is converted to electrical signals via chemical reactions. This process consumes high levels of oxygen and nutrients, making the retina one of the most metabolically active tissues in the human body. A well-organized ocular vascular system adapts to meet these metabolic requirements to ensure visual function. Hence, changes in the retinal vasculature serve as a biomarker for a number of ocular pathologies, such as advanced macular degeneration (Mullins et al., 2011) or glaucoma (Jesus et al., 2019), but also for other diseases such as hypertension (Klein et al., 1997) or diabetes (Fong et al., 2004).

A number of modalities have been developed for imaging the human retina over the last decades. Due to its simplicity and affordability, Fundus Photography (FP) has been extensively used, and it is nowadays present in most of the ophthalmic clinics. The fast acquisition and wide-field coloured images have kept FP relevant even when more complex/newer techniques, such as Optical Coherence Tomography (OCT), appeared. Although the vascular tree is usually clearly visible on FP images, changes in the retinal pigmentation, or lesions, may not be seen within the visible spectrum range. Hence, other *en face* imaging modalities have been developed, such as Scanning Laser Ophthalmoscopy (SLO). While in FP the image is obtained with one single shot, in SLO the laser beam scans the retina line by line. Although the acquisition time is slightly longer, the image quality is usually better than in FP. The increase in terms of quality or sharpness may not be clearly noticeable in healthy subjects, but has been reported in patients with diseases that affect the anterior chamber, such as cataract. In these cases, the evaluation of structures in scanning modalities has been described as more consistent (Kirkpatrick et al., 1995). Also, in diseases that only alter certain areas of the retina, such as geographic atrophy (Schmitz-Valckenberg et al., 2008), or reticular pseudodrusen (Schmitz-Valckenberg et al., 2011), an improvement has been observed. Another advantage of SLO is that this modality is usually integrated in OCT imaging devices. Thus, the findings of the SLO can be complemented with the OCT in-depth information. Lastly, unlike FP, SLO does not require pupil dilation to attain high-quality images, making it more comfortable for the patient (Kelly et al., 2003).

Regardless the imaging modality adopted in ophthalmic care, an accurate segmentation of the arteriovenous retinal tree is needed to support the clinical diagnosis and follow-up. However, to perform a manual segmentation is a tedious and a time-consuming task, specially if the capillaries are needed. Hence, extensive literature in vessel segmentation based on FP imaging (Srinidhi et al., 2017) has been published over the last two decades. Despite the number of procedures presented, conventional image processing has been replaced by convolutional neural network (CNN) based techniques over the last years. This is a trend observed not only on ophthalmic care but also in other medical fields of research. Therefore, a prior attention has been given to CNN in this work.

## 1.1. State-of-the-art

In (Wang et al., 2015), a CNN was used as feature extractor, combined with a random forest for classification. The architecture consisted of 6 layers, with input size $25 \times 25$. The approach was validated in two public databases, using $\sim$200k samples of each one. The input data was pre-processed. Similarly, (Fu et al., 2016) proposed a CNN in combination with a conditional random field, with the goal of creating an architecture specific for retinal vessel segmentation. An opposing point of view is explored in (Wu et al., 2016), where the authors' goal was to find a method widely applicable to diverse vessel tracking problems, not necessarily in the retina. The CNN is refined using principal components analysis (PCA). The approach is based on N4-fields (Ganin and Lempitsky, 2014), and it had a similar performance as the original N4-fields. In (Guo et al., 2018), an ensemble of networks was proposed, each of which had 10 layers and $64 \times 64$ inputs, without pre-processing. In (Liskowski and Krawiec, 2016), RGB patches of $27 \times 27 \times 3$ from three public datasets were used. They training set consisted of 3 to 5 millions of patches, depending on the dataset.

The authors analysed the effect of pooling, different types of pre-processing, and the use of data augmentation. They did not observe strong changes in these experiments, but they noticed an improvement when adding more patches on problematic/difficult regions. In (Oliveira et al., 2018), a new architecture was proposed. The authors applied both pre-processing and data augmentation. The data augmentation either did not improve the performance of the model or lead to a change in the dropout values or oversampling. The patch size was 88×88, and three public databases were used, taking between 2750 and 3750 patches per image depending on the image size in each dataset. In (Melinščak et al., 2015), a 10-layer architecture was proposed, without any pre-processing besides extracting the green channel. Some authors have also used pre-trained networks. In (Maninis et al., 2016), a pre-trained VGG was used, combining ideas from the Inception architecture to use feature maps of different sizes. In (Jiang et al., 2018), another transfer learning approach was proposed, based on the AlexNet architecture. The authors created patches of 50×50, and then resized them to 500×500 in order to enlarge the details. They applied pre-processing, and used more than 80000 patches for training. Moreover, they also focused on the post-processing to refine the outputs. In (Mo and Zhang, 2017), another pre-trained VGG was tested, similar to (Maninis et al., 2016). The validation was performed on three public datasets, and 5 to 10 patches were selected per image. Finally, there are some approaches such as (Girard et al., 2019), that combined vessel segmentation with specific applications, such as artery/vein classification. For the segmentation part, a U-net architecture was used. A median filter was applied in each channel of each patch, and then concatenated to the input, so that each input was composed by six channels. Data augmentation was also used.

In contrast with the amount of approaches that segmented blood vessels on FP, only one approach focused on SLO vessel segmentation was found in the literature. In (Meyer et al., 2017), the authors used a U-net (Ronneberger et al., 2015) trained on public datasets, and they achieved good results with patches of size 128×128.

From this succinct literature review, it can be concluded that most of the authors obtained comparable results with a wide range of different approaches. Although some works study the effect of pre-processing or data augmentation, most authors proposed specific architectures. Only (Meyer et al., 2017) for SLO and (Girard et al., 2019) for FP used a state-of-the-art network on image segmentation, U-net. The number of patches that has been reported changes from less than ten patches per image up to a few millions of patches in total. Such variability has also been observed for the size of the patches. Regarding validation, all the reviewed works used at least one of the publicly available datasets listed in Table 1, making the results of different approaches comparable.

Table 1: Publicly available datasets used in this work displayed by name, modality, number of images (# Im.), image size, and field-of-view (FoV).

| Name | Modality | # Im. | Image size | FoV |
|---|---|---|---|---|
| DRIVE (Staal et al., 2004) | FP | 40 | 584×565 | 45° |
| STARE (Hoover et al., 2000) | FP | 30 | 700×605 | 35° |
| HRF (Budai et al., 2013) | FP | 45 | 3304×2336 | 60° |
| CHASE_DB1 (Fraz et al., 2012) | FP | 28 | 1280×960 | 30° |
| IOSTAR (Zhang et al., 2016) | SLO | 30 | 1024×1024 | 45° |
| RC-SLO (Zhang et al., 2016) | SLO | 40 | 360×320 | 45° |

The main motivation for this work is to study if a model trained in one of the modalities, FP or SLO, can be used to segment the other accurately. This goal is tackled in two steps. First, a review on the existing approaches for vessel segmentation was performed in order to establish a basic pipeline. Consequently, a state-of-art segmentation architecture (U-net) was used, and the influence of two parameters, patch size and number of patches, was analysed. Taking into account the reviewed state-of-the-art, the pipeline was kept as simple as possible, without pre-processing, post-processing, or data augmentation. In the second part of this work, the pipeline was used to investigate the transferability of information between *en-face* retinal imaging modalities in vessel segmentation. The results for each modality individually, as well as the cross-modal evaluation, are presented and discussed.

## 2. Methods

Figure 1 depicts the pipeline adopted in this work. A fixed CNN architecture, U-net with Adam optimizer (learning rate 0.001) and 20% of dropout was trained during 1000 epochs. The U-net is a state-of-art architecture in medical image segmentation problems, and it got its name from its distribution in two branches: first, a contracting path, which applies a sequence of two 3×3 convolutions, ReLU, and 2×2 max pooling operations. The second branch is an expansive path, which consists in a sequence of upsampling, 2×2 convolution, two 3×3 convolutions, and ReLU operations. Additionally, there are connections between both branches of the network, in order to incorporate part of the feature maps from the contracting path in the computation of the expansive path. The last layer of the network performs a 1×1 convolution operation to map the components of the feature vectors into the desired number of classes. The CNN has a total of 23 convolutional layers.

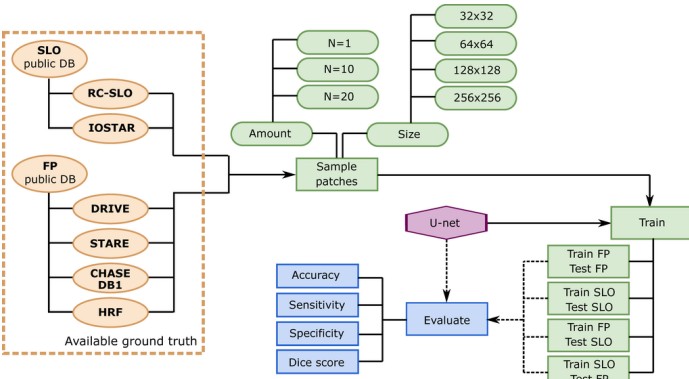

Figure 1: Schematics of the pipeline used in this work. N relates to the number of patches per image depending on the ratio between image and patch size.

The Dice score (Havaei et al., 2017) was used as a loss function for training the U-net CNN. This loss function was selected due to the nature of the labels, which are highly imbalanced, as the vessels represent only a small portion of the pixels in each image. The Dice score is expected to improve the sensitivity of the model, which is usually the least optimal parameter reported in literature (Srinidhi et al., 2017). Different patch sizes were

evaluated: 32, 64, 128, and 256. The smallest size was selected based on the literature review (Wang et al., 2015; Liskowski and Krawiec, 2016), and the largest size according to the smallest image within the datasets included (RC-SLO dataset). No pre-processing was considered, and only the green channel of the FP images, the most informative for vessel segmentation (Ramlugun et al., 2012; Staal et al., 2004), was used. No data augmentation was applied, as some previous works reported minimal improvement with the inclusion of augmentation (Liskowski and Krawiec, 2016; Oliveira et al., 2018) and accurate results without augmentation (Wang et al., 2015; Jiang et al., 2018).

In this work, six public datasets have been used, four in FP and two in SLO (see Table 1). All of the datasets have at least one manual segmentation available, and different image characteristics. For each dataset, 70% of the images were used for training, 20% for validation, and a final 10% for testing. The division was done at the image level instead of in the patches in order to not include patches for a given image in both training and validation sets, as some of them would overlap and bias the outcome. The specific number of patches per image depended on the image size and on the patch size, and it was computed as $(img_x \times img_y / patch_x \times patch_y) \times N$, where $(img_x \times img_y / patch_x \times patch_y)$ is the theoretical non-overlapping maximum number of patches that could be obtained per image. The factor $N$ varied between 1, 10, and 20. The batch size was as large as the available memory allowed for a specific patch size: 128, 64, 32, and 16, for the smallest to largest patch size respectively. For the test set, each pixel in the predicted image was classified in one of four categories: true positive, true negative, false positive, and false negative. Then, the accuracy, sensitivity, specificity, and Dice score were obtained.

## 3. Results

The inter-observer agreement was computed in the datasets that have several manual annotations available per image, in order to establish the maximum expected values for the evaluation metrics. The values of sensitivity, specificity, accuracy, and Dice were 0.81, 0.98, 0.96, and 0.79 for DRIVE; 0.64, 0.99, 0.95, and 0.74 for STARE; and 0.80, 0.98, 0.97, and 0.78 for CHASE_DB1. It can be observed that sensitivity and Dice score are quite low in comparison to accuracy and specificity, which is coherent with the outcomes reported in literature for automated segmentation methods (Appendix A, Table 4). These results are also justified by the nature of the data, as the most difficult part (for both manual and automated approaches) is to label the capillaries correctly.

The complete results for the U-net trained on each dataset using different patch sizes and number of patches are depicted in Table 2. A smaller N resulted on lower values for all the measures in most of the datasets. Only CHASE_DB1 and HRF datasets obtained a sensitivity comparable to the state-of-the-art. Although it was not the case for all datasets, a slight improvement was observed for N=20 compared to N=10 in terms of sensitivity, specificity, and accuracy, while the Dice score remained mostly unchanged. The results have also shown that larger patches provide better results regardless the dataset. Among all metrics, sensitivity and Dice score were the most affected by the patch size.

The configuration that obtained the best results overall (patch size = 256×256 and N = 20) was used for training and testing on the same dataset individually. The comparison between the predicted image for each dataset and the respective ground-truth is shown in

Table 2: Sensitivity, specificity, accuracy, and Dice score of each individual dataset for each # patches N and patch size. The highest values for each dataset are highlighted.

| Dataset | Size | N = 1 | | | | N = 10 | | | | N = 20 | | | |
|---|---|---|---|---|---|---|---|---|---|---|---|---|---|
| | | Sens. | Spec. | Acc. | Dice | Sens. | Spec. | Acc. | Dice | Sens. | Spec. | Acc. | Dice |
| DRIVE | 256 | 0.53 | 0.98 | 0.93 | **0.00** | 0.89 | **0.93** | **0.93** | 0.72 | 0.93 | **0.92** | **0.92** | **0.71** |
| | 128 | **0.63** | 0.97 | **0.94** | **0.00** | **0.92** | 0.92 | 0.92 | 0.68 | **0.96** | 0.91 | 0.91 | 0.67 |
| | 64 | 0.48 | 0.98 | 0.93 | **0.00** | 0.86 | 0.90 | 0.91 | 0.58 | 0.90 | 0.88 | 0.90 | 0.57 |
| | 32 | 0.00 | **1.00** | 0.91 | **0.00** | 0.66 | 0.90 | 0.91 | 0.42 | 0.68 | 0.88 | 0.90 | 0.41 |
| STARE | 256 | 0.04 | **1.00** | 0.96 | 0.00 | 0.82 | 0.97 | 0.96 | 0.69 | 0.78 | **0.97** | 0.95 | **0.69** |
| | 128 | **0.23** | 1.00 | 0.96 | 0.00 | 0.71 | **0.97** | 0.96 | 0.62 | **0.80** | 0.95 | 0.94 | 0.61 |
| | 64 | 0.13 | 1.00 | 0.96 | 0.00 | 0.56 | **0.97** | 0.96 | 0.45 | 0.60 | 0.96 | **0.95** | 0.45 |
| | 32 | 0.01 | 1.00 | 0.96 | 0.00 | 0.36 | **0.97** | 0.96 | 0.29 | 0.41 | 0.94 | 0.94 | 0.29 |
| HRF | 256 | **0.76** | 0.94 | 0.93 | **0.61** | 0.87 | 0.92 | **0.93** | 0.62 | 0.89 | 0.92 | **0.93** | 0.60 |
| | 128 | 0.63 | **0.95** | **0.94** | 0.50 | 0.73 | **0.93** | **0.93** | 0.52 | 0.76 | 0.91 | 0.92 | 0.49 |
| | 64 | 0.38 | **0.95** | **0.94** | 0.34 | 0.47 | 0.92 | 0.92 | 0.34 | 0.51 | 0.91 | 0.92 | 0.33 |
| | 32 | 0.23 | 0.92 | 0.92 | 0.17 | 0.29 | 0.88 | 0.89 | 0.20 | 0.31 | 0.87 | 0.89 | 0.20 |
| CHASE DB1 | 256 | **0.62** | **0.98** | **0.95** | 0.00 | **0.87** | 0.95 | 0.95 | 0.74 | **0.89** | 0.95 | 0.95 | **0.74** |
| | 128 | 0.57 | 0.97 | 0.94 | 0.02 | 0.76 | **0.95** | **0.95** | 0.59 | 0.79 | **0.95** | **0.95** | 0.59 |
| | 64 | 0.21 | **0.98** | **0.95** | 0.29 | 0.57 | **0.95** | **0.95** | 0.42 | 0.57 | 0.94 | 0.94 | 0.41 |
| | 32 | 0.17 | **0.98** | **0.95** | 0.05 | 0.36 | 0.92 | 0.93 | 0.24 | 0.34 | 0.92 | 0.93 | 0.24 |
| IOSTAR | 256 | 0.09 | **1.00** | 0.89 | 0.00 | **0.89** | 0.96 | 0.96 | **0.80** | 0.92 | 0.96 | 0.96 | 0.79 |
| | 128 | **0.65** | 0.98 | **0.95** | 0.00 | 0.83 | **0.96** | 0.95 | 0.71 | 0.86 | 0.95 | 0.95 | 0.71 |
| | 64 | 0.31 | 0.99 | 0.93 | **0.32** | 0.67 | 0.95 | 0.95 | 0.52 | 0.68 | 0.94 | 0.95 | 0.52 |
| | 32 | 0.17 | 0.99 | 0.93 | 0.25 | 0.40 | 0.93 | 0.94 | 0.30 | 0.42 | 0.92 | 0.93 | 0.30 |
| RC-SLO | 256 | **0.00** | **1.00** | 0.90 | 0.00 | 0.88 | 0.98 | 0.97 | 0.82 | 0.91 | 0.97 | 0.97 | **0.83** |
| | 128 | **0.00** | **1.00** | 0.88 | 0.00 | 0.84 | 0.98 | 0.96 | 0.81 | **0.91** | **0.97** | **0.97** | 0.79 |
| | 64 | **0.00** | **1.00** | 0.90 | 0.00 | 0.57 | **0.99** | 0.95 | 0.63 | 0.83 | 0.96 | 0.96 | 0.67 |
| | 32 | **0.00** | **1.00** | **0.91** | 0.00 | 0.48 | 0.96 | 0.96 | 0.41 | 0.52 | 0.95 | 0.95 | 0.40 |

Figure 2 (left). Figure 2 (right) depicts how the automatic segmentation is affected by varying the patch size.

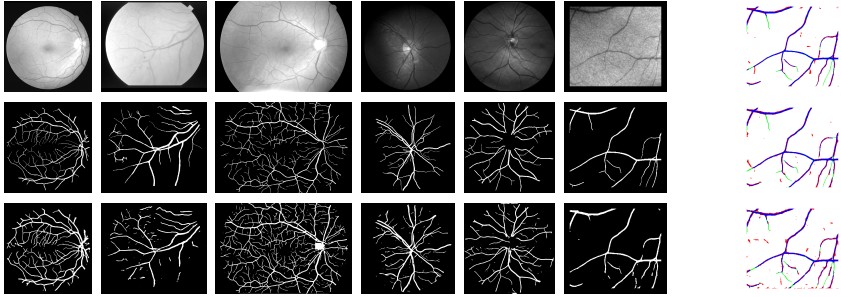

Figure 2: Left: retinal images (top), ground truth (middle), and output of the network (bottom), for datasets DRIVE, STARE, HRF, CHASE_DB1, IOSTAR, and RC-SLO. Right: evolution of true positives (blue), false positives (green) and false negatives (red) for patch size 128, 64, and 32 on the same RC-SLO image.

In order to observe if the results were transferable from one imaging modality to another, networks were trained and tested combining all the datasets of the same modality, as depicted in Table 3. Given the number of patches available for the FP modality is more than 3 times higher than the number of SLO patches, the amount of patches sampled per image

was lowered on the FP dataset to the same size of the SLO dataset. This way, the number of images for training was the same for both imaging modalities, avoiding an eventual bias due the different number of patches. As reference, the values for training and testing on the same type of image are also depicted. It can be observed that the outcome is the same than in Table 2, and the configuration with the largest patch size lead to the best performance. Regarding the training and testing in different modalities, it was observed that training in SLO and testing in FP yields significantly lower results, showing poor sensitivity and Dice. However, training in FP and testing in SLO yields accurate results, comparable to those obtained training and testing in the same modality.

Table 3: Sensitivity, specificity, accuracy, and Dice score for training and testing in the same and different modalities. All datasets were merged by modality. The highest values per dataset highlighted.

| Train set | Size | Test set FP | | | | Test set SLO | | | |
|---|---|---|---|---|---|---|---|---|---|
| | | Sens. | Spec. | Acc. | Dice | Sens. | Spec. | Acc. | Dice |
| FP | 256 | **0.93** | **0.90** | 0.91 | **0.61** | **0.94** | **0.93** | **0.93** | **0.71** |
| | 128 | 0.83 | **0.90** | **0.91** | 0.50 | 0.91 | 0.90 | 0.91 | 0.62 |
| | 64 | 0.62 | 0.89 | 0.90 | 0.37 | 0.74 | 0.90 | 0.91 | 0.48 |
| | 32 | 0.41 | 0.84 | 0.86 | 0.23 | 0.47 | 0.79 | 0.82 | 0.27 |
| SLO | 256 | 0.41 | **0.98** | 0.93 | 0.42 | 0.92 | 0.96 | 0.96 | 0.79 |
| | 128 | 0.33 | **0.98** | 0.93 | 0.32 | 0.88 | 0.95 | 0.95 | 0.71 |
| | 64 | 0.22 | **0.98** | 0.93 | 0.16 | 0.73 | 0.94 | 0.94 | 0.54 |
| | 32 | 0.15 | 0.97 | **0.93** | 0.11 | 0.45 | 0.92 | 0.93 | 0.32 |

## 4. Discussion

Automated segmentation has been a subject of study by the image processing community for quite some time, and the number of works based on deep learning have exponentially grown over the last decade, pushing the boundaries of what was possible in the domain of digital image processing. Challenging problems are now being solved with substantially better performance compared to traditional methods. This trend has also reached medical image processing, including ophthalmic research. Automated retinal vessel segmentation on FP have substantially improved since the introduction of CNNs. Besides the boost in performance, promising results have been shown in cross-training (training the network in one dataset and testing in a second dataset, with high variability in the image characteristics between both (Wang et al., 2015; Jiang et al., 2018)). In addition, the fact that these approaches are very fast (e.g. less than 1 second to process an image (Girard et al., 2019)), make them suitable for a real-time processing environment. Besides FP, good results have also been reported in SLO (Meyer et al., 2017). However, despite all research performed so far, it is difficult to identify an approach across the current options of architectures, components, pre- and post-processing, among other aspects that may influence the results, that could be easily applied and used in the clinical practice. Thus, in this work, a few guidelines that serve as a baseline for retinal vessel segmentation have been established.

In this work, it is shown that a simple CNN such as the U-net is good enough to replicate the current current results in the literature. These findings are in line with previous works (Isensee et al., 2018), that argue how a correctly tuned U-Net, also with a large patch size

and Dice score, can outperform more tailor-made approaches in brain segmentation. While this work does not show if data augmentation, pre-processing, or post-processing, could improve the results, the obtained accuracy, sensitivity, and specificity are on par with the values reported in the state-of-the-art. Moreover, the obtained values are also on the same range as the inter-observer agreement, and can be thus taken as theoretical maximum. The results show that the largest the input patch size, the better. However, the depth of the CNN is fixed in this work, and the smaller patches may suffer from issues at the deeper levels, such as insufficient resolution or border issues when down- and up-sampling. The size $256 \times 256$ provided the best results across all datasets. This also implies that a small number of images is enough to feed the network, as the largest patch sizes have been associated to a lower number of images. However, the number of patches must be sufficient, otherwise a dramatic drop on the sensitivity will occur, in agreement with the findings in (Oliveira et al., 2018). While the overall accuracy is barely affected, this metric must be handled carefully due to the imbalance of the labels. Overall, the main problem in all the reported methods is the sensitivity, as the capillaries tend to be ignored by the segmentation. Hence, this value should be always reported and carefully compared between methods. An interesting finding is that the results achieved for IOSTAR and RC-SLO in Table 2 have a higher sensitivity than the previously reported by (Meyer et al., 2017) in Table 4, despite of both approaches using a U-net. One of the key differences that may be causing the variation in the results is the choice of loss function, that in (Meyer et al., 2017) was the cross-entropy. In this work, Dice score was used, which emphasizes the weight of the true positives.

While many approaches have been proposed for FP, other imaging modalities, such as SLO, have not received that much attention. One of the causes for this lack of interest is the absence of public labeled datasets. Nevertheless, it is shown that the network trained in FP still provides an accurate segmentation on SLO data, but the opposite seems to not be true. Such results may be justified by the characteristics of both datasets. The FP datasets are more varied and have more pathological data. The difference on sharpness may also justify the performance between modalities. The fact that imaged vessels in FP may be less sharp than in SLO may make the algorithm more robust to different data. As a future work, augmentations in SLO imaging should be considered to infer whether a model trained on SLO could eventually be applicable to FP. Lastly, one should also consider the hypotheses that the green channel of the FP may be more informative than the infrared image acquired with SLO for segmenting the retinal vasculature.

## 5. Conclusion

In this work, it is shown that a state-of-the-art network, such as U-net, can be trained without pre-processing and augmentation and still perform as good as a manual grader, as far as a large patch size and enough images are used to train the CNN. The knowledge obtained from training on different modalities, such as FP or SLO, is transferable, but its sensitivity depends on the modality used to train the network. In this study, it is shown that, despite all its simplicity, a colour fundus photograph appears to be much more informative for training network than an image obtained from SLO. Hence, despite the lack of manual annotations on SLO images, coloured fundus photographs can be used to develop new and better networks with potential applicability in SLO imaging.

## Acknowledgments

This work was supported by the European research and innovation programme Horizon 2020 (grant agreement No 780989: Multi-modal, multi-scale retinal imaging project).

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

## Appendix A. Summary of state-of-art results on the public datasets

Table 4 summarizes the results reported by previous approaches on retinal vessel segmentation that propose a CNN, using (at least one of) sensitivity, specificity, and accuracy on the public datasets listed in Table 1.

Table 4: Values reported in previous works for deep learning-based vessel segmentation approaches. Best values for each dataset highlighted in bold.

| Work | Dataset | Sensitivity | Specificity | Accuracy |
|---|---|---|---|---|
| (Wang et al., 2015) | DRIVE | 0.8173 | 0.9733 | **0.9767** |
| | STARE | 0.8104 | 0.9791 | **0.9813** |
| (Fu et al., 2016) | DRIVE | 0.7294 | - | 0.9470 |
| | STARE | 0.7140 | - | 0.9545 |
| (Liskowski and Krawiec, 2016) | DRIVE | - | - | 0.9535 |
| | STARE | - | - | 0.9729 |
| (Guo et al., 2018) | DRIVE | **0.9859** | 0.7046 | 0.9613 |
| | STARE | **0.9861** | 0.5628 | 0.9539 |
| (Jiang et al., 2018) | DRIVE | 0.7540 | **0.9825** | 0.9624 |
| | STARE | 0.8352 | 0.9846 | 0.9734 |
| | CHASE_DB1 | **0.8640** | 0.9745 | **0.9668** |
| | HRF | **0.8010** | 0.8010 | **0.9650** |
| (Girard et al., 2019) | DRIVE | - | - | 0.9493 |
| (Oliveira et al., 2018) | DRIVE | 0.8405 | 0.9814 | 0.9639 |
| | STARE | 0.6329 | **0.9924** | 0.9365 |
| | CHASE_DB1 | 0.7731 | 0.9813 | 0.9600 |
| (Melinščak et al., 2015) | DRIVE | 0.7276 | - | 0.9466 |
| (Mo and Zhang, 2017) | DRIVE | 0.7779 | 0.9780 | 0.9521 |
| | STARE | 0.8147 | 0.9844 | 0.9674 |
| | CHASE_DB1 | 0.7661 | **0.9816** | 0.9599 |
| (Meyer et al., 2017) | IOSTAR | **0.8038** | **0.9801** | **0.9695** |
| | RC-SLO | **0.8090** | **0.9794** | **0.9623** |

