# OpenReview forum: "Deep learning-based retinal vessel segmentation with cross-modal evaluation"
_MIDL.io/2020/Conference — MIDL 2020_

### Official Review · AnonReviewer1 · 2020-02-19
**Reads well, good literature review, not clear what the main goal and finding are.**

**Rating:** 3
**Confidence:** 4
**Recommendation:** Poster

**Summary:**

The work looks into the use of deep learning for image segmentation of vessels from two retinal imaging modalities: fundus photography (FP) and scanning laser ophthalmology (SLO).

A U-net was trained, validated, and tested using six publicly available datasets, without data augmentation or pre-processing. Four of the datasets consisted of FP data, while the remaining two consisted of SLO data. Different combinations of training and testing were carried out, e.g. Train FP & Test FP; Train FP & Test SLO, etc.

Experiments were carried out using different settings in order to identify the effect of number of patches used for training, as well as chosen patch size, on segmentation performance.

**Strengths:**

Well written, interesting problem to explore, literature review covered a good number of papers. I found the use of different data source for training and testing particularly interesting, including the finding that training only on fundus photographs could result in good segmentation on SLO data.

**Weaknesses:**

1. It is not very clear what the main goal of the paper is. Is it to simply show that deep learning can work well for vessel segmentation? Is it to show that this can be done without data augmentation or pre-processing as mentioned several times? If so, the paper didn't show any experiments with data augmentation or pre-processing for comparison. Also, what advantages does not carrying out data augmentation could have? Perhaps a reduction in training time? Currently the paper shows that U-net works very well under different settings and on different data, but it is hard to grasp what the goal of the paper is.

2. The paper discussed how important DSC is, but did not actually report on DSC as an evaluation metric.

3. It is difficult to understand which set of experiments do the graphs in Figure 2 and 3 refer to, please clarify on the figures.

4. Key results are actually shown as tables in the appendices, which some readers might not refer to.

**Justification Of Rating:**

The work is interesting, the paper is well written, and a lot of experiments were carried out. However, it is not clear what the main goal of the experiments is, and how the goal relates to the findings.

**Paper Type:**

validation/application paper

**Questions To Address In The Rebuttal:**

What is the main goal of the paper, and how do the experimental findings relate to it? The abstract focuses on the ability to carry out segmentation without the need for data augmentation, whereas the conclusion focuses on findings like showing that a network can be trained on FP but still work well on SLO data.

**Special Issue:**

no

---

> ### Author Response · Authors · 2020-03-27
> **Rebuttal Reviewer 1**
>
> Thank you very much for your comments, please find below our answers to the different points and how we have addressed them in the paper.
>
> 1. It is not very clear what the main goal of the paper is. Is it to simply show that deep learning can work well for vessel segmentation? Is it to show that this can be done without data augmentation or pre-processing as mentioned several times? If so, the paper didn't show any experiments with data augmentation or pre-processing for comparison. Also, what advantages does not carrying out data augmentation could have? Perhaps a reduction in training time? Currently the paper shows that U-net works very well under different settings and on different data, but it is hard to grasp what the goal of the paper is.
> > The paper has two main goals, which are:
>     First, to review the existing approaches for vessel segmentation and establish a basic pipeline. To that end, a state-of-art segmentation architecture (U-net) and the influence of two parameters, patch size and number of patches, in the output, were analysed. The emphasis in the text regarding not using data augmentation and/or preprocessing is to highlight that the pipeline was kept as simple as possible. We are not implying that the results are better without augmentation, as we did not perform experiments on this aspect. However, authors that study the effect of augmentation for this particular problem in the past found that it did not have a strong impact on the model.
>     Second, to investigate the transferability of information between en face retinal imaging modalities in vessel segmentation. This is motivated by the lack of development and data available in SLO modality, in contrast with the larger datasets and numerous approaches for FP.
> The abstract was rewritten to highlight these two key ideas (please refer to the "Updated abstract" comment).
>
> 2. The paper discussed how important DSC is, but did not actually report on DSC as an evaluation metric.
> > As the labels are not balanced (i.e. there is way more background than vessels), we used a Dice-score based loss function, so that emphasis was put on the vessels (the under-represented class). The reason behind reporting accuracy, sensitivity, and specificity scores was for comparison purposes, as most of the authors in the past (both using conventional and deep learning methods) reported their findings using these parameters (for a better insight on this, please refer to the paper
> Srinidhi, C. L., Aparna, P., & Rajan, J. (2017). Recent advancements in retinal vessel segmentation. Journal of medical systems, 41(4), 70.
> which does an in-depth review on retinal vessel segmentation algorithms). However, we do agree with the Reviewer that Dice score should be reported, as it provides a fairer estimation of the model performance and, hence, we have added the values of test Dice score for all the experiments.
>
> 3. It is difficult to understand which set of experiments do the graphs in Figure 2 and 3 refer to, please clarify on the figures.
> 4. Key results are actually shown as tables in the appendices, which some readers might not refer to.
> > Reply to 3 & 4: It was also noted by other reviewers that the appendix tables were indeed more informative than the figures. Therefore, we have decided to move the tables to the main document, removing the plots (as the information was the same as in the tables).
>
> What is the main goal of the paper, and how do the experimental findings relate to it? The abstract focuses on the ability to carry out segmentation without the need for data augmentation, whereas the conclusion focuses on findings like showing that a network can be trained on FP but still work well on SLO data.
> > We have removed the mentions to augmentation and preprocessing from the abstract, as this was not the intended focus of our study, and we have clarified the main goals of this paper.

---

> > ### Comment · AnonReviewer1 · 2020-03-27
> > **Goal of the work clarified, reviewer's rating updated**
> >
> > Thank you for clarifying the goal of the work and for explaining how the points I raised will be addressed.
> >
> > My understanding from the rebuttal is that there are two main goals:
> > (1) To review retinal vessel segmentation approaches and to establish a baseline model, which involved studying the effects varying patch size and number of patches,
> > and (2)  To investigate transferability between FP and SLO, motivated by the apparent lack of SLO development and data available.
> >
> > I find both these goals interesting and I agree that literature review that was carried out, in addition to the experiments conducted, indeed go a long way to meeting these goals.
> >
> > However, I'm not sure if the new version of the abstract actually captures both these points. I suggest further rephrasing the abstract (and the paper) to clearly point out these two goals, as they still remain unclear to the reader. For instance, the new abstract does not mention anything about reviewing the literature. The abstract also moves quickly from the point on information transferability between the two modalities to the best observed choice of patch size.
> >
> > In other words, the goal and findings have now been made more clear to the reviewer, but they would very likely still remain unclear to the reader. I strongly suggest that the authors update the abstract and paper accordingly, as the work is actually quite interesting.
> >
> > Finally, given the changes that the authors mentioned they will carry out, and the clarifications on the goals and experimental design, I'm happy to revisit the rating given to the paper.
> >
> > ---------------------------------------------------
> > My updated rating: "Strong Accept".
> > ---------------------------------------------------

---

> > > ### Author Response · Authors · 2020-04-03
> > > **Updated abstract and goals**
> > >
> > > Thank you very much for your comments. We would like to share another version of the abstract and an updated paragraph in the introduction section. Please let us know your opinion regarding if they reflect the goals more clearly.
> > >
> > > Updated abstract:
> > >
> > > This work proposes a general pipeline for retinal vessel segmentation on en-face images. The main goal is to analyse if a model trained in one of two modalities, Fundus Photography (FP) or Scanning Laser Ophthalmoscopy (SLO), is transferable to the other modality accurately. This is motivated by the lack of development and data available in en-face imaging modalities other than FP. FP and SLO images of four and two publicly available datasets, respectively, were used. First, the current approaches were reviewed in order to define a basic pipeline for vessel segmentation. A state-of-art deep learning architecture (U-net) was used, and the effect of varying the patch size and number of patches was studied by training, validating, and testing on each dataset individually. Next, the model was trained in either FP or SLO images, using the available datasets for a given modality combined. Finally, the performance of each network was tested on the other modality. The models trained on each dataset showed a performance comparable to the state-of-the art and to the inter-rater reliability. Overall, the best performance was observed for the largest patch size (256) and the maximum number of overlapped images in each dataset, with a mean sensitivity, specificity, accuracy, and Dice score of 0.89$\pm$ 0.05, 0.95$\pm$0.02, 0.95$\pm$0.02, and 0.73$\pm$0.07, respectively. Models trained and tested on the same modality presented a sensitivity, specificity, and accuracy equal or higher than 0.9. The validation on a different modality has shown significantly better sensitivity and Dice on those trained on FP.
> > >
> > > Updated goals (last paragraph of Section 1):
> > >
> > > Given the variability in CNN architectures and parameters proposed over the last years, it is difficult to conclude on a general procedure that could be easily applied to different imaging modalities. The goal of this paper is two-fold: first, to propose a basic retinal vessel segmentation pipeline using a state-of-art architecture, U-net, analysing the influence that the parameters patch size and number of patches have in the output. Taking into account the findings in the literature, the pipeline was kept as simple as possible, without pre-processing, post-processing, or data augmentation. The second goal is to investigate the transferability of information between en face retinal imaging modalities in vessel segmentation. This is motivated by the wide range of approaches and larger datasets available on FP, while other en face imaging modalities, such as SLO, have seen less development.

---

> > > > ### Comment · AnonReviewer1 · 2020-04-03
> > > > **Reply to 'Updated abstract and goals'**
> > > >
> > > > Thank you for this. The new abstract now reads very well and is very clear. However, I don't understand why the updated goals in Section 1 is not consistent with the abstract. For example, the abstract seems to focus on the importance of studying transferability between FP and SLO as the main goal, while the updated text in Section 1 may suggest that this is not the focus of the paper as developing a pipeline is discussed first.
> > > >
> > > > Please ensure consistency throughout.

---

> > > > > ### Author Response · Authors · 2020-04-03
> > > > > **Updated goals**
> > > > >
> > > > > Thank you very much for the prompt and helpful reply. Indeed, the focus seemed different between the abstract and the goals, so we have rephrased the latter as follows:
> > > > >
> > > > > "The main motivation for this work is to study if a model trained in one of the modalities, FP or SLO, can be used to segment the other accurately. This goal is tackled in two steps. First, a review on the existing approaches for vessel segmentation was performed in order to establish a basic pipeline. Consequently, a state-of-art segmentation architecture (U-net) was used, and the influence of two parameters, patch size and number of patches, was analysed. Taking into account the reviewed state-of-the-art, the pipeline was kept as simple as possible, without pre-processing, post-processing, or data augmentation. In the second part of this work, the pipeline was used to investigate the transferability of information between en-face retinal imaging modalities in vessel segmentation. The results for each modality individually, as well as the cross-modal evaluation, are presented and discussed."

---

> ### Author Response · Authors · 2020-03-27
> **Updated abstract**
>
> "This work proposes a general pipeline for retinal vessel segmentation on en-face images, based on previous developments on Fundus Photography (FP), and studies its applicability to a different imaging modality, Scanning Laser Ophthalmoscopy (SLO). FP and SLO images of four and two publicly available datasets, respectively, were used in this study. First, the effect of varying the patch size and number of patches was studied in terms of accuracy, sensitivity, specificity, and Dice score. To that end, a U-net model was trained on each dataset individually, using 70% of the images for training, 20% for validation, and a final 10% for testing. Then, the model was trained in either FP or SLO images. The performance of each network was also tested on the other modality, in order to assess if the knowledge is transferable between them. The models trained on each dataset showed a performance comparable to the state-of-the art and to the inter-rater reliability. Overall, the best performance was observed for the largest patch size (256) and the maximum number of overlapped images in each dataset, with a mean sensitivity, specificity, accuracy, and Dice of 0.89$\pm$ 0.05, 0.95$\pm$0.02, 0.95$\pm$0.02, and 0.73$\pm$0.07, respectively. Models trained and tested on the same modality presented a sensitivity, specificity, and accuracy equal or higher than 0.9. The validation on a different modality has shown significantly better sensitivity and Dice on those trained on FP."

---

### Official Review · AnonReviewer4 · 2020-03-13
**Comparative analysis to generalize the deep learning based retinal vessel segmentation**

**Rating:** 3
**Confidence:** 5
**Recommendation:** Poster

**Summary:**

This paper aims to analyze the effect of the number of patches and respective sizes to the retinal vessel segmentation performance. To this end, comprehensive analysis has been performed using the U-net based framework to show its generalization ability. It also studies whether knowledge obtained from Fundus Photography (FP) image is transferable to another imaging modality, namely Scanning Laser Ophthalmoscopy (SLO) image. There are in total six public datasets that have been used for the validation and analysis.

**Strengths:**

1. The study about whether results were transferable from color fundus photography to SLO images is interesting.
2. The investigation of using varying patch sizes and the fusion of image of different datasets show the generalization ability of the framework.


**Weaknesses:**

1. The motivation and results are not explained clearly in Abstract and Introduction section.
2. Although extensive analysis and evaluations have been performed to show the practical values of this work, the methodology contribution is modest.


**Detailed Comments:**

1. Could the authors explain why only the U-net framework has been considered for the proof of concept. Have the authors also considered other latest state-of-the-art segmentation networks?
2. The citation to the IOSTAR and RC-SLO datasets seems not correct. Please revise it and refer to the correct paper below:
 J. Zhang, B. Dashtbozorg, E. Bekkers, J.P.W. Pluim, R. Duits, and B.M. ter Haar Romeny, “Robust retinal vessel segmentation via locally adaptive derivative frames in orientation scores,” IEEE Transactions on Medical Imaging, vol. 35, no. 12, pp. 2631–2644, 2016.
3. I would be beneficial if the authors also study how the established pipeline can be used to segment more challenging vessel structures such as small capillaries, strong central reflex and closely parallel vessels.


**Justification Of Rating:**

This paper presents a comprehensive analysis of the deep learning based retinal vessel segmentation framework on multiple image modalities. This paper is in general well-presented and easy to follow. The comparative analysis shows some interesting aspects. However the investigations can still be better improved by including more specific studies on important issues like the segmentation on challenging vessel structures.

**Paper Type:**

validation/application paper

**Questions To Address In The Rebuttal:**

1. The incorrect citation needs to be fixed. See detailed comments.
2. More detailed examples of qualitative segmentation comparisons could be included to better present the performance of different network configurations. For example, if one choice gives better performance, in which aspect does it really improve?

**Special Issue:**

no

---

> ### Author Response · Authors · 2020-03-27
> **Rebuttal Reviewer 4**
>
> Thank you very much for your comments, please find below our answers to the different points and how we have addressed them in the paper.
>
> 1. The motivation and results are not explained clearly in Abstract and Introduction section.
> > The abstract has been rewritten to clearly state the goals of the paper:
> "This work proposes a general pipeline for retinal vessel segmentation on en-face images, based on previous developments on Fundus Photography (FP), and studies its applicability to a different imaging modality, Scanning Laser Ophthalmoscopy (SLO). FP and SLO images of four and two publicly available datasets, respectively, were used in this study. First, the effect of varying the patch size and number of patches was studied in terms of accuracy, sensitivity, specificity, and Dice score. To that end, a U-net model was trained on each dataset individually, using 70% of the images for training, 20% for validation, and a final 10% for testing. Then, the model was trained in either FP or SLO images. The performance of each network was also tested on the other modality, in order to assess if the knowledge is transferable between them. The models trained on each dataset showed a performance comparable to the state-of-the art and to the inter-rater reliability. Overall, the best performance was observed for the largest patch size (256) and the maximum number of overlapped images in each dataset, with a mean sensitivity, specificity, accuracy, and Dice of 0.89$\pm$ 0.05, 0.95$\pm$0.02, 0.95$\pm$0.02, and 0.73$\pm$0.07, respectively. Models trained and tested on the same modality presented a sensitivity, specificity, and accuracy equal or higher than 0.9. The validation on a different modality has shown significantly better sensitivity and Dice on those trained on FP."
>
> 2. Although extensive analysis and evaluations have been performed to show the practical values of this work, the methodology contribution is modest.
>
> 1. The incorrect citation needs to be fixed. See detailed comments.
> > The correct citation was added as suggested.
>
> 2. More detailed examples of qualitative segmentation comparisons could be included to better present the performance of different network configurations. For example, if one choice gives better performance, in which aspect does it really improve?
> > In order to reflect this remark, Figure 3 has been changed, adding for one dataset the comparison between output and manual label varying the patch size.
>
> 1. Could the authors explain why only the U-net framework has been considered for the proof of concept. Have the authors also considered other latest state-of-the-art segmentation networks?
> > The U-net was chosen due to its simplicity and versatility, as it has been shown to perform accurately in a wide variety of medical image segmentation scenarios. Besides, other authors have successfully used a U-net in the past to demonstrate same performance than more tailor-made models, as illustrated in the paper:
> Isensee, F., Kickingereder, P., Wick, W., Bendszus, M., & Maier-Hein, K. H. (2018, September). No new-net. In International MICCAI Brainlesion Workshop (pp. 234-244). Springer, Cham.
> If we restrict the literature review to the field of retinal vessel segmentation, there is not a clear state-of-the-art network, as most authors define their own architectures.
>
> 2. The citation to the IOSTAR and RC-SLO datasets seems not correct. Please revise it and refer to the correct paper below:
>        J. Zhang, B. Dashtbozorg, E. Bekkers, J.P.W. Pluim, R. Duits, and B.M. ter Haar Romeny, “Robust retinal vessel segmentation via locally adaptive derivative frames in orientation scores,” IEEE Transactions on Medical Imaging, vol. 35, no. 12, pp. 2631–2644, 2016.
> > Thank you for pointing out this mistake. The reference has been added.
>
> 3. It would be beneficial if the authors also study how the established pipeline can be used to segment more challenging vessel structures such as small capillaries, strong central reflex and closely parallel vessels.
> > These are indeed some of the main challenges on retinal vessel segmentation, and potential future lines of work.

---

> > ### Author Response · Authors · 2020-04-03
> > **Updated Abstract**
> >
> > Please find below an updated version of the abstract. We hope that it offers a more clear view on the goals of the paper.
> >
> > This work proposes a general pipeline for retinal vessel segmentation on en-face images. The main goal is to analyse if a model trained in one of two modalities, Fundus Photography (FP) or Scanning Laser Ophthalmoscopy (SLO), is transferable to the other modality accurately. This is motivated by the lack of development and data available in en-face imaging modalities other than FP. FP and SLO images of four and two publicly available datasets, respectively, were used. First, the current approaches were reviewed in order to define a basic pipeline for vessel segmentation. A state-of-art deep learning architecture (U-net) was used, and the effect of varying the patch size and number of patches was studied by training, validating, and testing on each dataset individually. Next, the model was trained in either FP or SLO images, using the available datasets for a given modality combined. Finally, the performance of each network was tested on the other modality. The models trained on each dataset showed a performance comparable to the state-of-the art and to the inter-rater reliability. Overall, the best performance was observed for the largest patch size (256) and the maximum number of overlapped images in each dataset, with a mean sensitivity, specificity, accuracy, and Dice score of 0.89$\pm$ 0.05, 0.95$\pm$0.02, 0.95$\pm$0.02, and 0.73$\pm$0.07, respectively. Models trained and tested on the same modality presented a sensitivity, specificity, and accuracy equal or higher than 0.9. The validation on a different modality has shown significantly better sensitivity and Dice on those trained on FP.

---

### Official Review · AnonReviewer3 · 2020-03-13
**Interesting but incomplete experimental paper, misleading title**

**Rating:** 1
**Confidence:** 5

**Summary:**

The authors train a standard architecture on several datasets of fundus and SLO retinal images and report performance for each of these experiments and also cross-modality testing. Special emphasis is given to the impact of patch size used for training the models. Experiments show the somehow interesting fact that training on SLO is not a very good idea if the model is to be used for standard fundus images.

**Strengths:**

- Few papers (maybe none) have reported experimental analysis on training in one modality and testing in another without re-training. It is interesting to know that training on fundus and testing on SLO may be ok, but not the other way round.
- The analysis on patch size is also a good point of the paper.
- A good amount of datasets is considered in the experimental section, more than usual.

**Weaknesses:**

- When I first read the title I thought the authors were going to propose a method that would learn to segment retinal blood vessels simultaneously from different modalities, given that it has the words "cross-modal learning" on it. However, that is not the topic of the paper, but rather what happens is that they have "cross-modal evaluation", which is a very different thing.
- Numerical results are relegated to the appendix, and the paper's experimental section is left as quite weak in my opinion. No single numerical result is present in the main paper, only two sets of graphs that show the evolution of accuracy, sensitivity and specificity as a function of patch size. In my opinion, it is not fair to tell the reader that s/he should go to the appendix to see the interesting part of the paper, which is how well does this method work on the different datasets.
- I don't think it is correct to rely only on accuracy, sensitivity and specificity at a given threshold used to binarize the predictions of a CNN. Area under the ROC curve should be reported, and also f1-score or some other metric that does not depend so much on class imbalance (accuracy is rather useless in this problem, it will be super-high for almost every method you consider).
- I believe at the very least some baseline performances of other methods should have been reported together with what is in the paper now, just to know how does this compare with current state-of-the-art (I understand it will not be the bets method one can find, but that is no reason not to report some comparison).

**Justification Of Rating:**

The experimental evaluation is not very well designed and reported in this paper, which makes it hard to understand the conclusions. Also, without looking in the appendix it is impossible to know about the actual outcome of reading this paper. In addition, I don't think there is much relevance/novelty to what is proposed in this paper.  Maybe the authors could consider doing actual cross-modality learning by learning in some way jointly from both modalities. If a method trained like that would outperform methods trained only on SLO or only on Fundus images, that would be more interesting!

**Paper Type:**

validation/application paper

**Questions To Address In The Rebuttal:**

- Please modify the title so that it does not induce to error.
- Figures 2 and 3 are hardly readable, and do not add so much of interest to the experimental section. Also figure 3 has lots of lost space on the sides, due to these graphs being so "tall". I believe there is the need to remove some of these results from this section, and make room for actual numerical results like the ones given in the appendix. In other words, I find more informative what the authors report in the appendix than what they actually report in the main paper.
- I would advice to add performance measurements in terms of AUC and F1 scores, and possibly remove accuracy or at the very least use balanced accuracy* .
- It seems to me that the last part of section 2 is not about methodology but about results, isn't it?

* https://scikit-learn.org/stable/modules/generated/sklearn.metrics.balanced_accuracy_score.html

**Special Issue:**

no

---

> ### Author Response · Authors · 2020-03-27
> **Rebuttal Reviewer 3**
>
> Thank you very much for your comments, please find below our answers to the different points and how we have addressed them in the paper.
>
> When I first read the title I thought the authors were going to propose a method that would learn to segment retinal blood vessels simultaneously from different modalities, given that it has the words "cross-modal learning" on it. However, that is not the topic of the paper, but rather what happens is that they have "cross-modal evaluation", which is a very different thing.
> > The title has been changed to fit the content of the paper accurately. The new title is:
> Deep learning-based retinal vessel segmentation with cross-modal evaluation
>
> Numerical results are relegated to the appendix, and the paper's experimental section is left as quite weak in my opinion. No single numerical result is present in the main paper, only two sets of graphs that show the evolution of accuracy, sensitivity and specificity as a function of patch size. In my opinion, it is not fair to tell the reader that s/he should go to the appendix to see the interesting part of the paper, which is how well does this method work on the different datasets.
> > The results were relegated to the appendix due to the number of pages/space restrictions. However, we do agree with this point (that is shared through all the reviewers) that the numeric results need to be included within the paper, and not the appendix. Therefore, we have moved the tables from the appendix to the main text, substituting the figures.
>
> I don't think it is correct to rely only on accuracy, sensitivity and specificity at a given threshold used to binarize the predictions of a CNN. Area under the ROC curve should be reported, and also f1-score or some other metric that does not depend so much on class imbalance (accuracy is rather useless in this problem, it will be super-high for almost every method you consider).
> > We fully agree with the reviewer that accuracy is not informative in this type of problem (this was also pointed out in the Discussion section), and that sensitivity and specificity, or AUC, give a better perspective. In fact, the problem of data imbalance is discussed when explaining why the Dice score was chosen as the loss function. The only purpose of reporting accuracy is for comparison with previous works, as almost all the authors provide accuracy, sensitivity, and specificity, while very few use additional metrics, such as Dice. Nevertheless, we have decided to add the Dice score to the results, as it reflects better the performance of the model.
>
> I believe at the very least some baseline performances of other methods should have been reported together with what is in the paper now, just to know how does this compare with current state-of-the-art (I understand it will not be the bets method one can find, but that is no reason not to report some comparison).
> > A table with baseline performances is available in the appendix, but could not be fit into the main document due to the page limit.
>
> Please modify the title so that it does not induce to error.
> > The title has been changed to reflect the contents more accurately.
>
> Figures 2 and 3 are hardly readable, and do not add so much of interest to the experimental section. Also figure 3 has lots of lost space on the sides, due to these graphs being so "tall". I believe there is the need to remove some of these results from this section, and make room for actual numerical results like the ones given in the appendix. In other words, I find more informative what the authors report in the appendix than what they actually report in the main paper.
> > Following this suggestion, we have moved the results tables from the appendix to the main document, in substitution of the figures.
>
> I would advice to add performance measurements in terms of AUC and F1 scores, and possibly remove accuracy or at the very least use balanced accuracy* .
> > The measures sensitivity, specificity, and accuracy, were selected because they are reported in most of the works in retinal vessel segmentation, hence enabling comparison with previous studies, as the goal of the paper was not to improve the state of the art, but to study if a well-known and widely used model, such as the U-net, would have similar results to the many specific architectures that have been proposed over the years.
> However, we agree with the reviewers that measures such as AUC, F1, or Dice score (as suggested by the other reviewers), which are robust to class imbalance, are much more informative, given the nature of the data. Therefore, we have added the Dice score in all the experiments, and also for the inter-rater comparison.
>
> It seems to me that the last part of section 2 is not about methodology but about results, isn't it?
> > The inter-observer agreement for the datasets that have more than one manual evaluation was moved from the Methods section to the start of the Results section.

---

### Meta-Review · Area_Chair1 · 2020-04-06
**MetaReview of Paper249 by AreaChair1**

**Rating:** 3
**Recommendation For Accepted Papers:** Poster

**Metareview:**

This paper proposes a deep learning-based method for retinal vessel segmentation based on cross-modal learning.
The rebuttal convinced most of the reviewers.

**Paper Type:**

validation/application paper

**Special Issue:**

no

---

### Decision · Program_Chairs · 2020-04-11

Accept